# Coordinated spinal locomotor network dynamics emerge from cell-type-specific connectivity patterns

F David Wandler[1], Benjamin K Lemberger[1], David L McLean[2], James M Murray[1]*

[1]Institute of Neuroscience, University of Oregon, Eugene, United States; [2]Discovery Brain Sciences, University of Edinburgh, Edinburgh, United Kingdom

## eLife Assessment

In this **valuable** study, Wandler et al. provide **convincing** theoretical evidence for alternate mechanisms of rhythm generation by CPGs. Their model shows that cell-type-specific connectivity and an inhibitory drive could underlie rhythm generation. Excitatory input could act to enhance the frequency range of these rhythms. This modeling study could motivate further experimental investigation of these mechanisms to understand CPG rhythmogenesis.

*For correspondence:
jmurray9@uoregon.edu

Competing interest: The authors declare that no competing interests exist.

**Abstract** Even without detailed instruction from the brain, spinal locomotor circuitry generates coordinated behavior characterized by left–right alternation, segment-to-segment propagation, and variable-speed control. While existing models have emphasized the contributions of cellular- and network-level properties, the core mechanisms underlying rhythmogenesis remain incompletely understood. Further, neither family of models has fully accounted for recent experimental results in zebrafish and other organisms pointing to the importance of cell-type-specific intersegmental connectivity patterns and recruitment of speed-selective subpopulations of interneurons. Informed by these experimental findings and others, we developed a hierarchy of increasingly detailed models of the locomotor network. We find that coordinated locomotion emerges in an inhibition-dominated network in which connectivity is determined by intersegmental phase relationships among interneurons and variable-speed control is implemented by recruitment of speed-selective subpopulations. Further, while structured excitatory connections are not necessary for rhythmogenesis, they are useful for increasing peak locomotion frequency, albeit at the cost of smooth transitions at intermediate frequencies, suggesting a basic computational trade-off between speed and control. Together, this family of models shows that network-level interactions are sufficient to generate coordinated, variable-speed locomotion, providing new interpretations of intersegmental excitatory and inhibitory connectivity, as well as the basic, recruitment-based mechanism of speed control.

## Introduction

It has long been known that the spinal cord is capable of generating locomotor movements even in the absence of detailed instructive input from the brain (*Brown and Sherrington, 1911*). The core dynamical features of spinal locomotor circuitry are left–right alternation and rostro-caudal propagation of cyclical activity over a range of frequencies. These core features are most obvious during swimming in aquatic vertebrates (*Grillner and Wallén, 1985*), but they are also apparent during limbed locomotion (*Bonnot et al., 2002*; *Yakovenko et al., 2002*; *Cazalets, 2005*; *Ivanenko et al., 2006*; *Cuellar et al., 2009*; *Saltiel et al., 2016*).

Despite decades of work, the mechanisms by which rhythm is generated for coordinated locomotion are not yet fully understood. One line of research has focused on the intrinsic properties of individual excitatory spinal interneurons, whose mutual interactions serve to orchestrate rhythmic oscillations among downstream interneurons and motor neurons (*Grillner and Kozlov, 2021*; *Rancic and Gosgnach, 2021*; *Grillner and El Manira, 2015*). This perspective is supported by the observation of rhythmic bursting of excitatory interneurons in response to tonic input (*Song et al., 2020*; *Agha et al., 2024*), but it is potentially challenged by the observation that targeted disruptions of sources of phasic inhibition interfere with rhythmogenesis (*Buchanan and McPherson, 1995*; *Moult et al., 2013*; *Picton et al., 2022*; *Agha et al., 2024*). Another line of work has suggested that rhythmic oscillations may be generated as a network-level phenomenon via random recurrent connectivity, without requiring specialized single-cell properties (*Lindén et al., 2022*). While this model is supported by the observation of 'rotational' population dynamics rather than purely biphasic dynamics (*Lindén et al., 2022*), the possible roles of particular cell types and their characteristic connectivity patterns are not addressed. In addition, the observation of recruitment of speed-selective interneuron populations (*McLean and Fetcho, 2011*; *Grillner and El Manira, 2015*) and the functional roles of cell-type-specific intersegmental connectivity within spinal cord (*Sengupta and Bagnall, 2023*) have not yet been fully accounted for in either the cellular- or the network-level perspective.

Here, we construct a hierarchy of increasingly detailed models in which spinal central pattern generation occurs in a distributed manner, and the essential ingredients for producing coordinated locomotion are cell-type- and speed-specific connectivity motifs rather than specialized cellular properties or random connectivity. In all of these models, we represent individual neurons in a highly simplified way, so the dynamical properties of the network are entirely driven by emergent population dynamics due to connectivity patterns among units. First, we show that letting connectivity be determined by intersegmental phase relationships in a single-population, inhibition-dominated network is sufficient to account for left–right alternation and rostro-caudal propagation. Next, we show that dividing our units into fast- and slow-preferring populations is sufficient to account for variable-frequency control with constant phase lag via speed-dependent recruitment. Finally, in our most detailed model, we show that the strength and modularity of recurrent excitation facilitates faster locomotion, but that there is a trade-off between maximum speed and controllability at intermediate speeds.

Together, these results point to an updated model of the spinal locomotor network as a distributed pattern generator, in which rhythm generation and other locomotion features emerge from collective population dynamics via patterned connectivity and speed-dependent recruitment.

## Results

To investigate the degree to which population-specific connectivity patterns might account for the phenomenology of the spinal network described above, we developed a family of models at different levels of detail. In order to focus our investigation on connectivity patterns and the emergent population dynamics that result from them, we modeled the dynamics of individual neurons in a highly simplified way:

$$\tau_i \frac{dr_i(t)}{dt} = -r_i(t) + \left[ \sum_j \boldsymbol{W}_{ij} \boldsymbol{r}_j(t - \Delta_{ij}) + D_i \right]_+ , \tag{1}$$

where $r_i(t)$ is interpreted as the firing rate of unit $i$ at time $t$, $\tau_i$ is the membrane time constant, $D_i$ is the tonic drive, $W_{ij}$ are recurrent weights from other units in the circuit, and $[.]_+$ denotes rectification. The axonal time delays $\Delta_{ij}$ are proportional to the number of segments between units $i$ and $j$.

The three subsections that follow describe a hierarchy of increasingly detailed models, all of which are described by this basic dynamical equation. This approach enables us to focus mainly on basic computational properties within the relatively abstract, high-level model, whereas the lower-level models focus on implementation of distinct cell-type populations for obtaining additional features.

### Phase relationships determine connectivity in a distributed pattern generator

Recent work mapping cell-type-specific connectivity patterns in zebrafish (*Menelaou et al., 2014*; *Sengupta et al., 2025*; *Sengupta and Bagnall, 2023*), mouse (*Alaynick et al., 2011*; *Goulding, 2009*),

and other organisms has revealed a substantial presence of both excitatory and inhibitory connectivity traversing multiple segments. This is somewhat difficult to interpret in light of classical models of the spinal locomotor circuit, in which dynamical single-cell properties such as bursting or synaptic fatigue generate oscillations within each segment, and these oscillators are then coupled together with intersegmental interactions (see *Cohen et al., 1992*; *Sigvardt and Miller, 1998*; *Grillner et al., 2007*; *Ausborn et al., 2021* for reviews). We hypothesized that spatially patterned connectivity alone may be sufficient to drive coordinated locomotion even in the absence of dynamical single-cell properties. To test this, we constructed a high-level model in which dynamical single-cell properties are absent, and the basic features of left–right alternation and segment-to-segment propagation are driven by connectivity patterns. We took inspiration from 'moving bump' models of brain circuits, for example in models of head-direction circuits, which have shown that inhibition-dominated, asymmetric connectivity can give rise to activation sequences (*Zhang, 1996*; *Samsonovich and McNaughton, 1997*; *Murray and Escola, 2017*). Such models are also motivated by recent work in zebrafish demonstrating the importance of strong phasic inhibition for generating rhythmic locomotor activity (*Agha et al., 2024*). This high-level model featured a single, homogeneous population of units described by *Equation 1*, where all units have identical membrane time constant $\tau$ and common excitatory input drive $D$.

The connections $W_{ij}$ between units in this model were all set to be negative or zero, so that units in the model receive excitation only from the tonic drive, not from each other. Previous related work on threshold-linear recurrent networks with binary inhibitory weights and tonic input has shown that, despite their relative simplicity, such networks are capable of supporting rich dynamics (*Curto and Morrison, 2023*). Due to the (piecewise) linearity of the activation function, the magnitude of the nonzero weights has no effect on the dynamics, so we set $W_{ij} = -1$.

Given these simplifying choices, the nontrivial question in designing this model is which pairs of units should inhibit one another. The key principle that this model led us to is that the connectivity between pairs of units should encode the desired phase relationships between those units. A version of this idea is already present in classical models, where left and right populations of neurons within

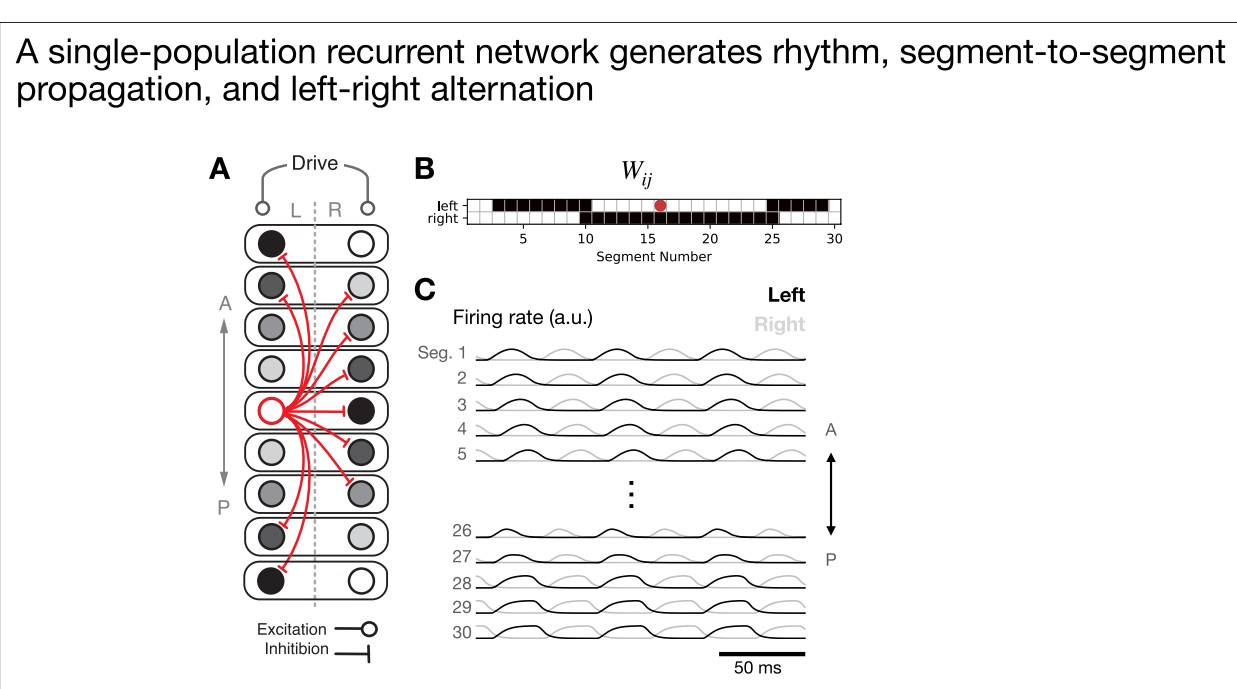

## A single-population recurrent network generates rhythm, segment-to-segment propagation, and left-right alternation

**Figure 1.** A single-population recurrent network generates rhythm, segment-to-segment propagation, and left–right alternation. (**A**) Schematic connectivity diagram. Grayscale circles represent individual units at different phases of oscillation, with one unit per hemi-segment. Red lines schematically illustrate inhibitory projections from a mid-body unit. (**B**) Weight matrix illustrating all outgoing inhibitory projections from the mid-body unit labeled with a red dot. (**C**) Time dependence of network activity with all units receiving the same tonic drive. The different traces represent different units.

each segment laterally inhibit one another, ensuring that they fire exactly out of phase. Here, we apply this same principle to model the longitudinal connectivity of the spinal locomotor network.

According to this principle, a mid-body unit should inhibit contralateral units in the same segment and in nearby segments, as well as inhibiting ipsilateral units more distally (shown schematically for a $N = 9$-segment model in *Figure 1A*). To make this precise, we can summarize the desired phase relationships for a pair of units $i$ and $j$ as follows:

$$\phi_{ij} = \frac{(s_i - s_j) \bmod N}{N} + \frac{(a_i - a_j) \bmod 2}{2},$$ (2)

where $N$ is the number of segments. In this equation, $s_i$ denotes the segment number of unit $i$, and $a_i = 0$ if unit $i$ is in the left hemi-segment or 1 if it is in the right hemi-segment.

In order to realize these desired phase relationships, we set the connection strength from neuron $j$ to neuron $i$ as $W_{ij} = f(\phi_{ij})$, where $\phi_{ij} \in [0, 1)$ is calculated from *Equation 2*, and

$$f(\phi) = \begin{cases} -1, & \phi_l < \phi < \phi_u, \\ 0, & \text{else.} \end{cases}$$ (3)

This equation describes local contralateral inhibition together with more-distal ipsilateral inhibition. Because $\phi_{ij} = 0.5$ for two units that are out of phase, it is necessary to choose $\phi_l < 0.5$ and $\phi_u > 0.5$. Further, in order for propagation to proceed head-to-tail rather than tail-to-head, it is necessary for the connectivity to be asymmetric, with a relatively larger window of disinhibition downstream and a relatively smaller window of disinhibition upstream. Hence, we chose $\phi_l = 0.3$ and $\phi_u = 0.8$ to define the window of inhibition. Finally, we can note that the most-distal ipsilateral projections described by *Equation 3* (of length $\gtrsim N/2$) are not actually necessary, since the units targeted by such inhibition already receive contralateral inhibition from nearby active units on the opposite side. Indeed, such long-range projections are not typically found in the zebrafish (*Higashijima et al., 2004*; *Callahan et al., 2019*; *Sengupta et al., 2021*). Hence, in this model and in the others presented below, all connections between segments more than 13 segments apart were set to 0 (*Figure 1B*).

Driving this network with tonic input, and setting the axonal delay $\Delta_{ij} = 0$ for simplicity, we observed the key features of coordinated locomotion: phasic bumps of activity in each segment, with smooth propagation of the bump from head to tail and strict alternation between left and right within each segment (*Figure 1C*). While this high-level model is a highly simplified and abstract representation of the spinal locomotor network, it illustrates several key ideas that may be relevant for characterizing locomotor circuitry in biological organisms and will continue to be present in the more elaborate models developed in the following subsections. First, rhythm can be generated by the network as a whole, rather than by dynamical single-neuron properties or single-segment oscillations. Second, the network's recurrent connectivity is dominated by inhibition, illustrating that inhibition is capable of and perhaps necessary for sculpting the dynamics. Finally, the pattern of inhibitory connectivity is determined by the desired phase relationships between pairs of units. In the following sections, we will build upon this model by introducing multiple interneuron populations and allowing for more-heterogeneous connectivity between units.

## Fast and slow speed modules implement frequency control via recruitment

While the single-population model described above was able to implement coordinated locomotion, it did so at a single, fixed frequency. This frequency was set by the only timescale in the model: the membrane time constant. Inspired by observations of *speed-module* structure in spinal circuitry in zebrafish (*McLean et al., 2008*) and mouse (*Rancic et al., 2020*), whereby interneurons are recruited at either fast or slow locomotion speeds but not both, we made a minimal change to our initial model that introduces an additional timescale, replacing each unit by two units: a 'fast' unit with a 1-ms membrane time constant, and a 'slow' unit with a 10-ms time constant, as observed experimentally in larval zebrafish (*Menelaou et al., 2022*). Trivially, in the absence of connectivity between the fast and slow modules, this model amounts to two copies of our earlier model and is capable of operating at two speeds: a fast speed if tonic input is provided only to the fast-preferring units, and a slow speed

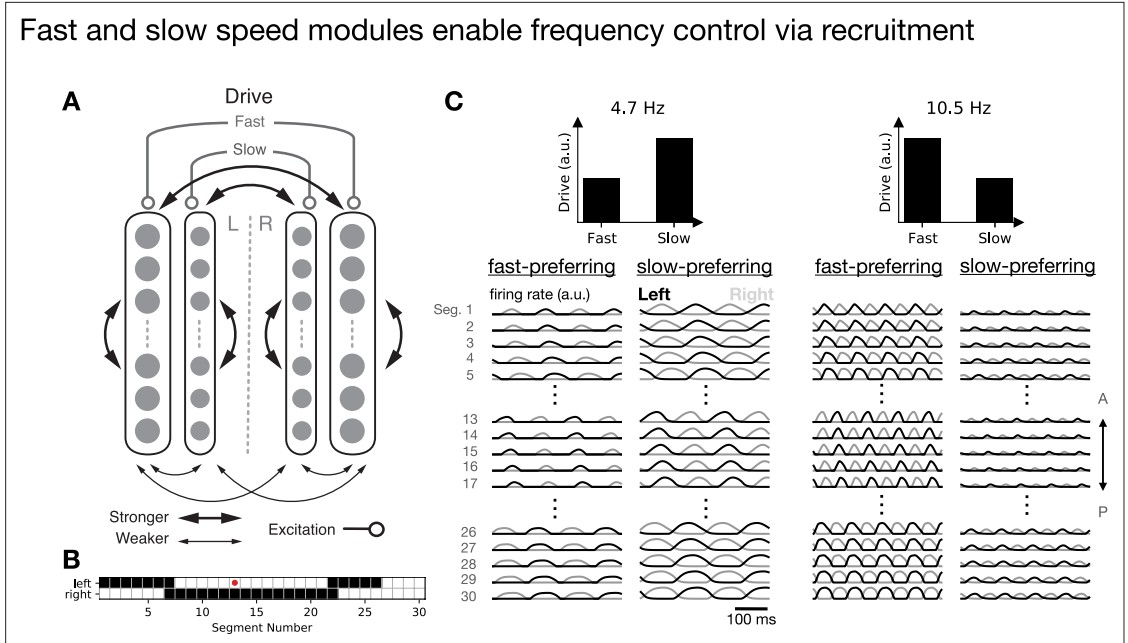

**Figure 2.** Fast and slow speed modules enable control of locomotion frequency. (**A**) Connectivity schematic illustrating that fast- and slow-module units receive distinct tonic drives and project both within and between modules. (**B**) Map of outgoing projections from a mid-body unit (red). (**C**) Time-dependent activity of fast- and slow-module units given different levels of tonic drive to the two populations.

if tonic input is provided only to the slow-preferring units (***Pujala and Koyama, 2019***). In each case, the period of left–right alternation and head-to-tail propagation would be roughly proportional to the corresponding membrane time constant (though perhaps not perfectly proportional if axonal delays are included).

In order for the two parts of the network to oscillate at a single global frequency and to interpolate continuously between slow and fast speeds, connectivity between the two modules is required. To begin, we adopted the simplest choice and made the connectivity weights the same for between-module projections and within-module projections, with the same spatial connectivity patterns for between- and within-module projections (***Figure 2A, B***). We found that the circuit exhibited coordinated locomotion characterized by head-to-tail propagation and left–right alternation at a single global frequency that was intermediate between the characteristic frequencies of the fast and slow modules. Further, the locomotion frequency could be controlled in a graded manner by driving the two modules with different inputs. Locomotion was fast when the fast units received most of the tonic drive, and it was slow when the slow units received most of the tonic drive, with smooth interpolation of intermediate frequencies as the ratio of tonic drives was varied (***Figure 2C***).

Analyzing the model in greater detail revealed that varying the relative levels of tonic drive to the two populations strongly modulated the frequency (***Figure 3A, B***). This is made possible by selective recruitment of fast and slow speed modules, such that the fast population is active at faster locomotion frequencies, while the slow population is active at slower locomotion frequencies, with a smooth crossover between these two regimes (***Figure 3C***). Such recruitment has been observed experimentally (***McLean et al., 2008***) but has not been accounted for by previous models. In ***Figure 3—figure supplement 1***, we further show that the disengagement of the slow module as frequency increases can occur due to inhibition from the increasingly active fast module, without requiring changes in the tonic drive to the slow population, which is also observed experimentally (***McLean et al., 2008***).

To establish that the model exhibits coordinated locomotion across all frequencies, we additionally computed the left–right phase difference between the pairs of units on either side of each segment, finding that the phase differed by half of a period across all locomotion frequencies (***Figure 3D***). We then computed the phase difference between pairs of units in adjacent segments, finding that this phase was approximately equal to $1/N$ across all locomotion frequencies, where $N$ is the number of segments (***Figure 3E***), so that the length of the spinal cord exhibits approximately one period of

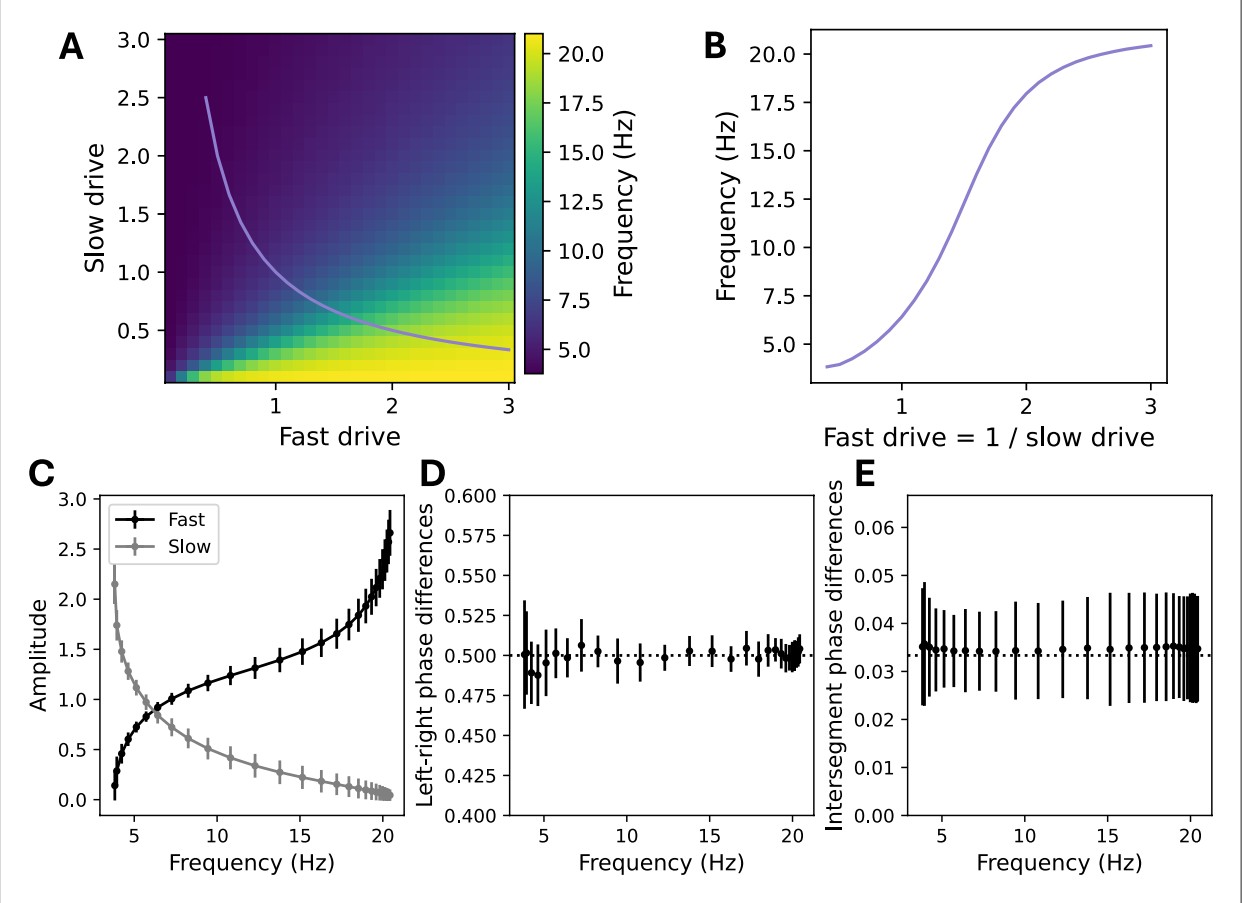

**Figure 3.** Speed-module recruitment enables frequency control. (**A**) Locomotion frequency with different levels of tonic drive to fast and slow units. (**B**) Frequency along the path shown in (**A**). (**C**) Frequency-dependent recruitment of fast and slow units as a function of locomotion frequency along the path shown in (**A**). (**D**) Phase difference between left and right units within each segment (dotted line corresponds to half of a period). (**E**) Phase difference (where 1 corresponds to a full period of oscillation) between pairs of units on the same side in adjacent segments (dotted line corresponds to $1/N$, where $N = 30$ is the number of segments). Error bars in all panels denote standard deviation across units.

The online version of this article includes the following figure supplement(s) for figure 3:

**Figure supplement 1.** Recruitment of the fast module at high frequencies inhibits the slow module.

**Figure supplement 2.** Frequency determination from time series is performed through calculating the period from the autocorrelation spectrum.

oscillation. This *constant phase lag* relationship is a core feature of locomotion that has been observed during swimming in a variety of animals (*Skinner and Mulloney, 1998*). While it does not appear by default in models that chain individually oscillating segments together via longitudinal excitation, constant phase lag emerges in our model as a natural consequence of the fact that the desired phase relationships between segments are built into the circuitry via long-range inhibitory projections. Because the phase relationships are determined by the connectivity, which is the same at all speeds, the phase relationships do not depend on locomotion frequency.

These results show that, by including fast and slow subpopulations and coupling these sub-networks together, the model succeeds in producing coordinated locomotion across a broad range of frequencies, with control of frequency via selective recruitment of the two subpopulations. While this model succeeds in producing much of the phenomenology of the spinal locomotor network, it does not yet fully address the diversity of excitatory and inhibitory cell types that are known to play a role in the biological circuitry (*Sengupta and Bagnall, 2023*). We turn our attention to this question in the following section.

## Excitatory and inhibitory cell types

To address the role of cell-type-specific connectivity patterns in greater detail, we next incorporated excitatory and distinct inhibitory populations by replacing each unit in the above model with four

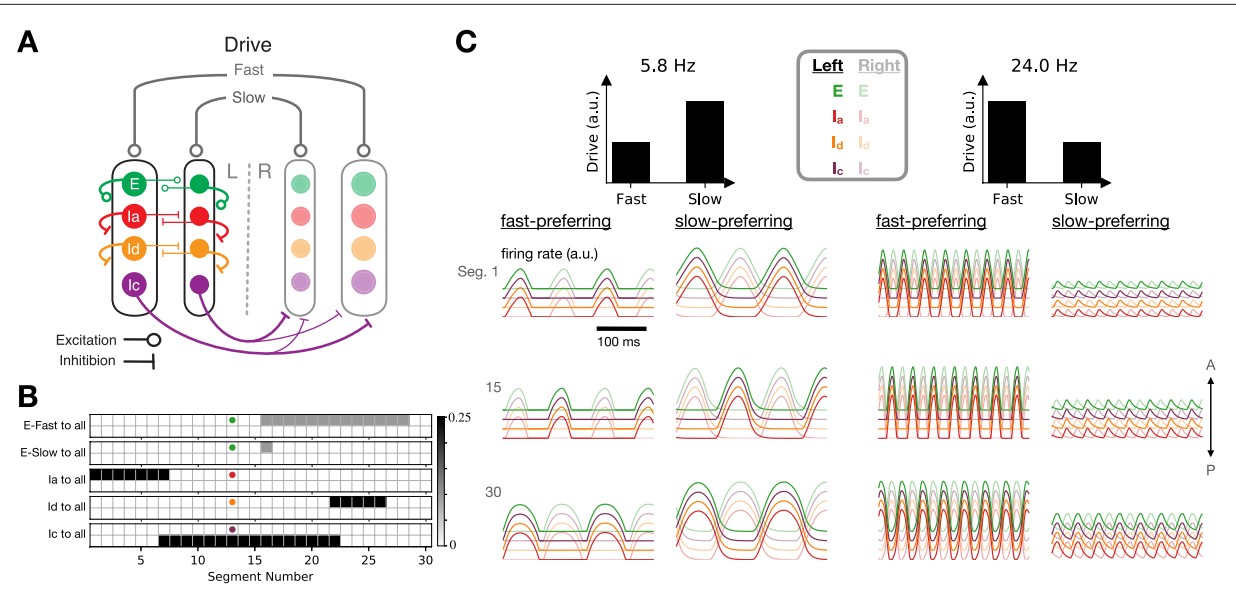

**Figure 4.** A model with excitatory and inhibitory populations. (**A**) Schematic diagram illustrating connectivity among cell types (but not longitudinal connectivity) for the eight-population model. (**B**) Detailed connectivity matrices for an example mid-body unit from each population. (**C**) Time-dependent activity traces at slow (left) and fast (right) locomotion frequencies (traces are slightly offset for clarity).

The online version of this article includes the following figure supplement(s) for figure 4:

**Figure supplement 1.** Speed-module recruitment enables coordinated locomotion with frequency and amplitude control in an eight-population model.

units, leading to an eight-population model (since each of the four types has fast and slow subtypes) (*Figure 4A*). One of these populations (corresponding to V2a interneurons in the zebrafish) consisted of excitatory units with descending, ipsilateral projections. The other three populations consisted of inhibitory units, essentially breaking the inhibitory population from the earlier model into three populations that have identical activity (since they all receive the same inputs) but differ in their projection targets. One population of inhibitory units (corresponding to V1 interneurons in the zebrafish) had ascending, ipsilateral projections; another (corresponding to V2b interneurons in the zebrafish) had descending, ipsilateral projections; and a third (corresponding to dI6 and V0d interneurons in the zebrafish) had contralateral projections. The tonic drive was provided equally to all units.

The outgoing spatial connectivity of each of these cell types is illustrated in *Figure 4B*. Each cell-type projects equally to all of the units within each segment that it targets, so that all units within each segment receive the same inputs. As in the two-population model above, we set the spatial connectivity patterns for the inhibitory units according to the desired phase relationships between units, with short-range contralateral inhibition and intermediate-range ipsilateral inhibition. For the excitatory units, we assumed that the projections are descending only in order to facilitate head-to-tail propagation. All units within each speed module had the same membrane time constant and axonal conduction velocity. As in the earlier models, providing distinct tonic drives to the fast- and slow-module units led to coordinated locomotion at a range of frequencies (*Figure 4C*).

Before analyzing the full model in detail, we decoupled the two speed modules from one another and began by studying the effects of various single-cell and cell-type-specific connectivity properties on the characteristic oscillation frequency of an individual speed module. Unsurprisingly, the locomotion frequency depended strongly on the membrane time constants, with smaller values of these parameters leading to faster frequencies (*Figure 5A, B*). Because the characteristic frequencies of the two speed modules set the upper and lower limits of locomotion frequency once the modules are coupled together in the full model, it is likely advantageous for an organism to have values of these parameters that differ strongly in fast- and slow-preferring neurons. This agrees with observations from the zebrafish, where the membrane time constants and axonal delays differ for fast- and slow-preferring excitatory and inhibitory interneurons (*Menelaou and McLean, 2019*; *Menelaou et al.,*

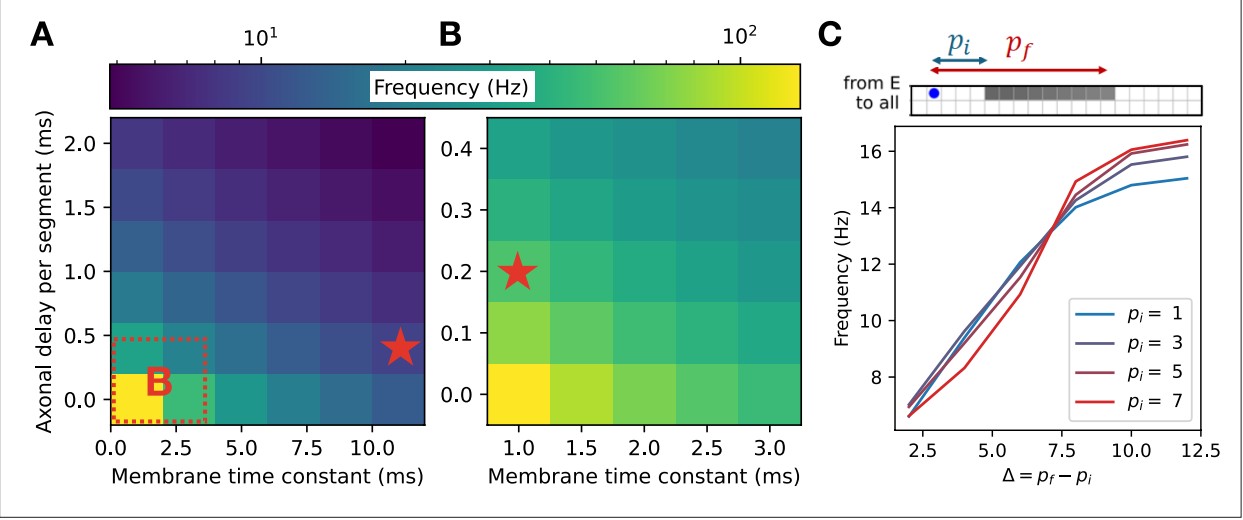

**Figure 5.** Single-cell properties and excitatory connectivity influence locomotor frequency in an individual speed module. (**A, B**) Dependence of locomotion frequency on the axonal delay per segment and membrane time constant of units ((**A**) shows a broad range of values; (**B**) shows an inset from (**A**)). Stars denote experimentally observed values for fast and slow excitatory V2a cells in zebrafish (*Menelaou and McLean, 2019*; *Menelaou et al., 2022*). (**C**) Dependence of locomotion frequency on the projection distances of excitatory connections originating from the excitatory unit labeled blue.

*2022*). For our subsequent simulations, we fixed these parameters for the fast and slow modules at the experimentally determined values indicated in *Figure 5A, B*; *Menelaou and McLean, 2019*; *Menelaou et al., 2022*.

We next investigated the effect of connectivity properties on locomotion frequency for the decoupled speed module. We found that the frequency was modulated by more than a factor of two as the excitatory projection distances were varied (*Figure 5C*). This agrees with observations from zebrafish, where the extents of intersegmental projections have been shown to differ for fast- and slow-preferring excitatory interneurons, with fast-preferring V2a interneurons projecting more distally than slow-preferring V2a neurons (*Menelaou et al., 2014*). For our subsequent simulations, we fixed these parameters for the fast and slow modules at the experimentally determined values illustrated in *Figure 5B* (*Menelaou et al., 2014*).

Having shown that the spatial extent of excitatory projections has a strong effect on locomotion frequency, we next asked whether varying connectivity properties would also modulate the range of possible frequencies in the full model with two coupled speed modules. Varying the global strength of excitatory projections had a strong effect on the range of possible frequencies, with stronger excitation facilitating faster locomotion (*Figure 6A*). In particular, whereas the purely inhibitory model with experimentally determined membrane time constants and axonal conduction velocities realizes a maximum frequency much lower than that observed in larval zebrafish (approximately 20 Hz, black line in *Figure 6A*), the inclusion of excitatory interneurons facilitates maximum frequencies of over 50 Hz, which is approaching peak swim speeds in larval zebrafish (*Agha et al., 2024*). Thus, while excitatory interneurons are not necessary for producing coordinated locomotion in our model, they do facilitate faster locomotion, suggesting that this may be a fundamental role for feedforward excitation in the spinal network.

Given that connectivity within and between speed modules has been shown in zebrafish to be modular, with stronger projections within modules than between modules (*Song et al., 2020*), we asked what would be the effect of varying modularity in the model. We defined modularity as the difference between intra- versus inter-module connection strength divided by the sum of these quantities, such that modularity of 1 corresponds to fully decoupled modules, while modularity of 0 corresponds to identical connection strengths within versus between modules.

Varying the modularity of all four populations together, we found that there was essentially no effect on the maximum or minimum possible frequencies. Further, the model lost the ability to produce locomotion at intermediate frequencies as modularity was increased (*Figure 6B*). However,

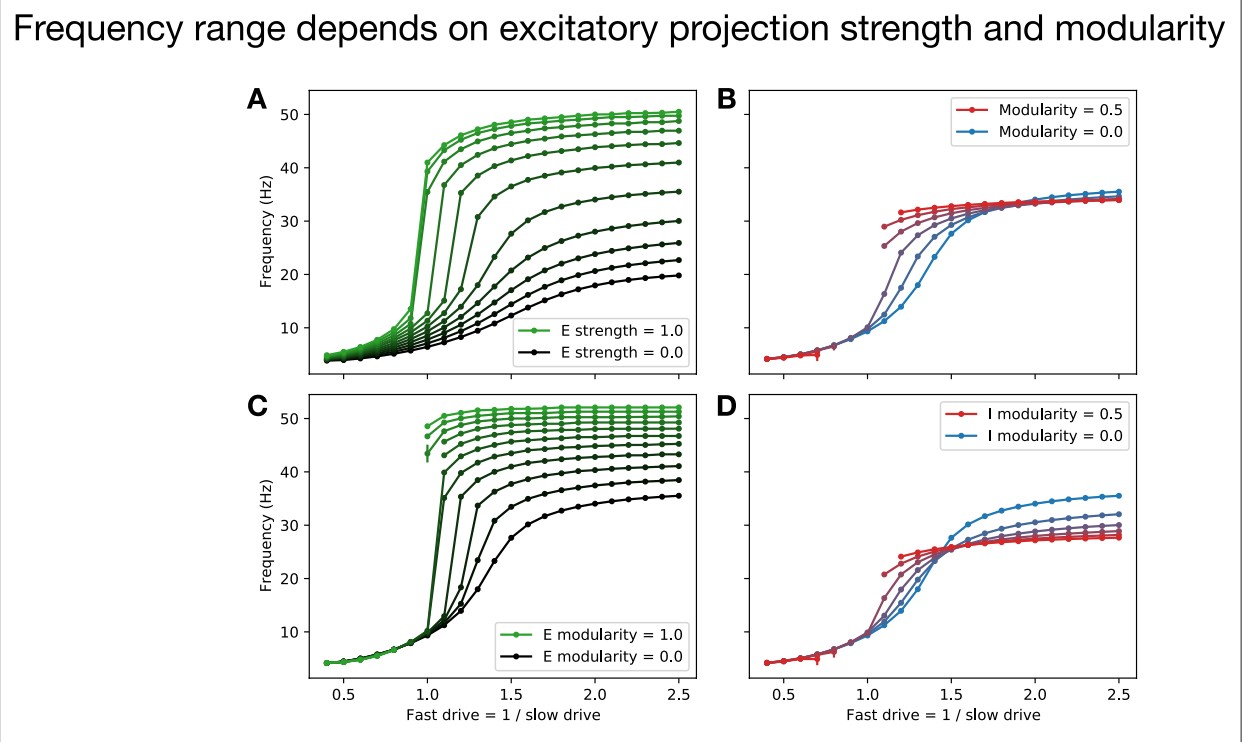

**Figure 6.** Frequency range depends on excitatory projection strength and modularity. (**A**) Dependence of the range of possible locomotion frequencies on the global strength of excitatory projections relative to that of inhibitory projections. (**B**) Dependence of the frequency range on connectivity modularity, which quantifies the strength of inter-module (fast-to-slow and slow-to-fast) projections relative to intra-module (fast-to-fast and slow-to-slow) projections. (Missing intermediate points correspond to cases where coordinated locomotion does not appear.) (**C**) Dependence of the frequency range on connectivity modularity of excitatory units, where inhibitory units have modularity set to zero. (**D**) Dependence of the frequency range on connectivity modularity of inhibitory units, where excitatory units have modularity set to zero. (In (**A**), modularity is set to zero; in (**B-D**), strength of excitation is set to 0.4.)

when we varied modularity among only the excitatory or only the inhibitory populations, we observed much more significant changes in the maximum frequency (*Figure 6C, D*). These changes occurred in opposite directions, with excitatory (inhibitory) modularity favoring faster (slower) speeds, suggesting that the lack of an observed change in frequency range when both types of modularity were varied together (*Figure 6B*) was due to cancellation between these two effects.

Together, these results show that the strength of feedforward excitation and the modularity of excitatory connectivity have a strong effect on the range of possible locomotion frequencies. There is a trade-off, however, in that the model loses the ability to smoothly interpolate between fast and slow frequencies in cases where the excitatory connectivity becomes too strong or too modular (*Figure 6A, C*). This requirement that excitation not be too strong is in accord with experimental observations from zebrafish (*Agha et al., 2024*), which have shown that peak excitatory post-synaptic currents are much weaker than peak inhibitory post-synaptic currents in V2a interneurons, consistent with the possibility that excitation may be globally weaker than inhibition in the spinal circuitry. Further, the fact that the model exhibits a frequency range similar to that of the zebrafish for parameters that are close to the critical values where smooth frequency control becomes impossible suggests that the spinal locomotor circuit faces a trade-off between speed and controllability, and that its excitatory connectivity may be configured in a way that optimizes this trade-off.

Having established the roles played by single-cell and connectivity properties of different cell types in the eight-population model, we fixed these parameters and analyzed the behavior of the model over the range of possible tonic drives to fast and slow populations (*Figure 4—figure supplement 1*). Similar to the two-population model, the eight-population with coupled fast and slow speed modules model exhibited head-to-tail propagation with constant phase lag (*Figure 4—figure supplement 1G*), left–right alternation (*Figure 4—figure supplement 1F*), and frequency-dependent recruitment

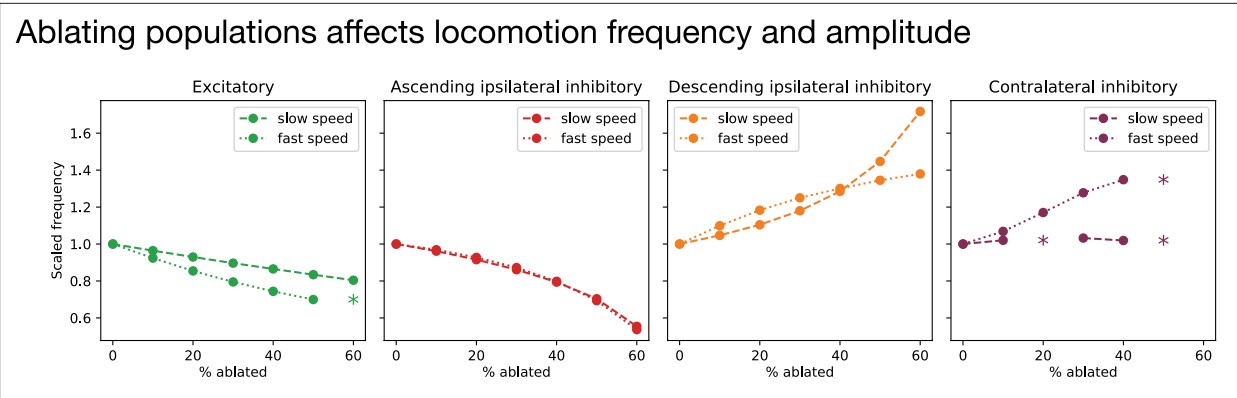

**Figure 7.** Ablating populations affects locomotion frequency dependence of locomotion frequency (normalized to its unperturbed value) on ablation of each of the four interneuron populations during slow speed oscillations (dashed lines; fast drive = 1.0, slow drive = 1.0; frequency = 9.3 Hz) and fast speed oscillations (dotted lines; fast drive = 2.0, slow drive = 0.5; frequency = 34.0 Hz). Asterisks mark points where the model failed to produce a coherent oscillation (see Methods).

of fast and slow populations (*Figure 4—figure supplement 1E*). In addition to varying the frequency of oscillations, we also found that the overall amplitude of interneuron activity in the model could be varied by co-varying the drives to the fast and slow populations (*Figure 4—figure supplement 1C, D*). This provides a potential mechanism to independently control frequency and amplitude of loco-motion, although the manner in which the amplitude of interneuron activity relates to the amplitude of locomotion would depend on the assumptions made about how interneuron activity drives the activity of motor neurons, which we have not included in our models.

We next investigated the effects of perturbing the model by partially ablating (i.e. attenuating the outgoing activity of) each interneuron population (*Figure 7*). At all locomotion speeds, we found that ablating excitatory units decreased locomotion frequency. This is in agreement with experiments in zebrafish, where ablation of excitatory V2a interneurons had the same effect (*Eklöf-Ljunggren et al., 2012*). Further, we found that ablating inhibitory units with ascending ipsilateral projections decreased locomotor frequency, while ablating inhibitory units with descending projections increased locomotor frequency across all locomotion speeds. This is also in agreement with experiments in zebrafish, where ablation of inhibitory V1 interneurons slowed swimming (*Kimura and Higashijima, 2019*), while abla-tion of V2b interneurons led to faster swimming (*Callahan et al., 2019*; *Sengupta et al., 2025*). Finally, we found that ablating the contralaterally projecting inhibitory units led to a modest increase in frequency, but that coordinated locomotion was lost when the degree of ablation became too great. The impact on frequency was most obvious at fast speeds, with a more modest impact at slow speeds. This is consistent with recent experiments in zebrafish, which found the impact of attenuating contralateral inhibitory projections from dl6 neurons on coordination was most obvious at fast speeds (*Agha et al., 2024*). Similar results were found in *Xenopus*, where silencing contralaterally projecting inhibitory interneurons can eliminate rhythm generation (*Moult et al., 2013*) or lead to an increase in swim frequency (*Dale, 1995*). Together, these results show that, where comparisons with experimental data are possible, perturbations to our model lead to effects on locomotion frequency that generally agree with experimental observations. This agreement provides support for the possibility that the basic mechanisms underlying variable-frequency locomotion in our model—namely cell-type-specific connectivity patterns and speed-module recruitment—may also be at play in the spinal locomotor network.

## Robustness in a biophysical model

One potential pitfall with the 'rate models' considered above is that the oscillatory behavior seen in these models might be overwhelmed by the inherent stochasticity of a biophysical setting. To test the robustness of our connectivity-based mechanisms for rhythmogenesis and frequency control, we built a spiking-neuron version of each model.

The spiking models were generated from the rate models by replacing each unit in the rate model with $n$ leaky-integrate-and-fire (LIF) neurons. A rough estimate suggests that the locomotor circuit is

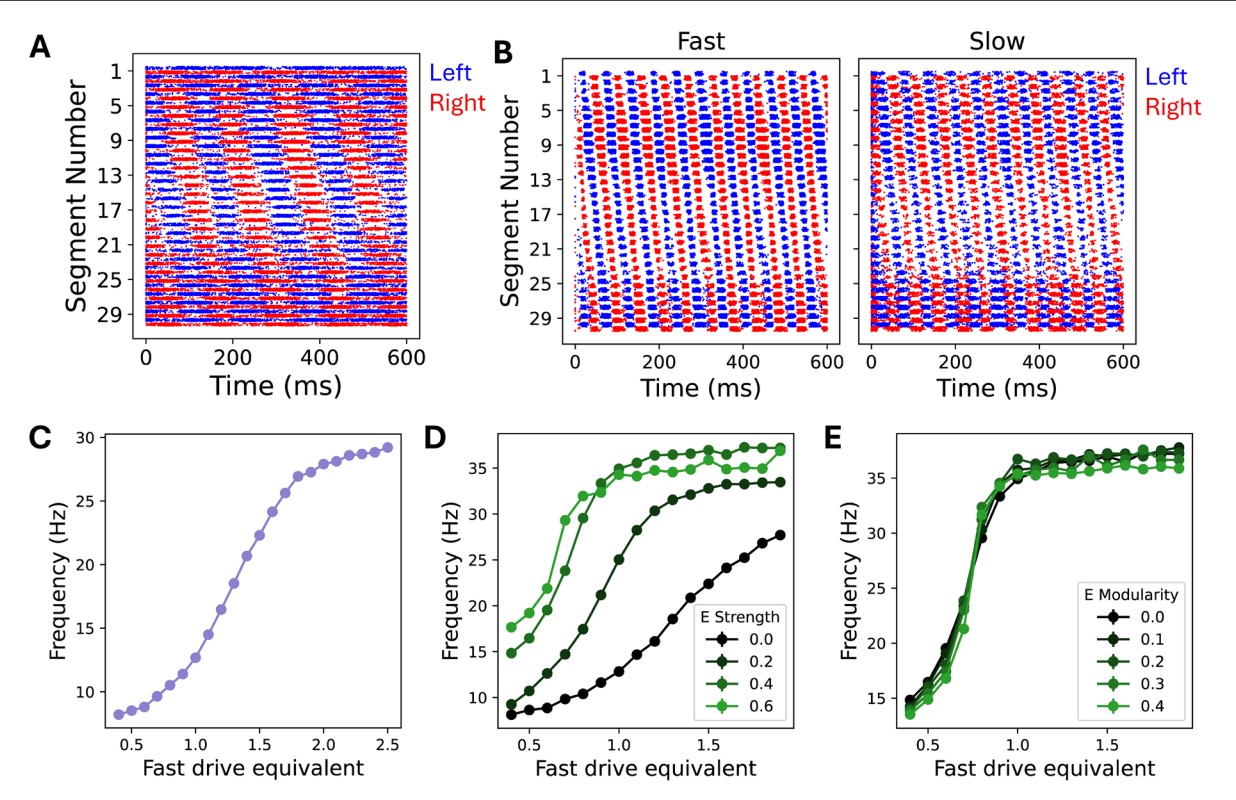

**Figure 8.** A spiking network model confirms that network-based mechanisms for rhythmogenesis and frequency control are robust in stochastic settings. (**A**) A raster plot from the single-population spiking model. The blue (red) dots show spikes from neurons on the left (right) side of the spinal cord. (**B**) A raster plot from the two-population spiking model, with spikes from the fast (slow) population shown in the left (right) panel. (**C**) Dependence of the frequency in the two-population spiking model on the driving rate. Given a fast drive equivalent, $f$, the EI-balanced Poisson inputs have rates chosen to produce $f \times 100$ Hz in an isolated fast neuron and $1/f \times 100$ Hz in an isolated slow neuron. (**D**) Dependence of the frequency in the eight-population spiking model on the strength of excitatory connections relative to the strength of inhibitory connections. The simulations shown here use 0 modularity for all populations. (**E**) Dependence of the firing rate oscillations in the eight-population spiking model on the modularity of the excitatory connections. The simulation uses an excitatory connection strength of 0.4 and 0 modularity for the inhibitory connections. All spiking model simulations shown here use $n = 80$ leaky-integrate-and-fire (LIF) neurons for each unit of the corresponding rate model and connectivity fraction $p = 0.1$.

The online version of this article includes the following figure supplement(s) for figure 8:

**Figure supplement 1.** Dependence of spiking model behavior on $n$, $p$, and $\tau$.

made up of 50–100 neurons of each cell type per segment (*Agha et al., 2024*; *Callahan et al., 2019*; *Sengupta et al., 2021*; *Kishore et al., 2020*). In accordance with this estimate, most of our simulations use $n = 80$, though versions with smaller $n$ also performed well (*Figure 8—figure supplement 1*). Connectivity between neurons in the spiking model was determined by including a random fraction $p$ of all possible connections between groups of neurons corresponding to connected units in the rate model. The strength of these connections was determined by the strength of the corresponding connection in the rate model (for details, see Methods).

In addition to the stochasticity introduced via the randomized connectivity, we introduced stochasticity in the spiking model by replacing the tonic input drive of the rate model with EI-balanced Poisson input. This input was independently sampled for each neuron, with a uniform rate for each speed population. In the rate model, a given tonic drive represented the firing rate that an isolated unit would maintain without recurrent connections. To similarly control the input to the spiking model, the rates of the EI-balanced Poisson inputs were chosen to produce a target firing rate in an isolated LIF neuron. Since frequency control depended primarily on the ratio of the fast and slow tonic drives, and the target firing needed to be high enough to ensure spikes would occur reliably within the maxima of the oscillations, we chose our target rates to be 100 Hz times the equivalent tonic drive.

The spiking version of the single-population model demonstrated left–right alternation and head-to-tail propagation (*Figure 8A*). As with the rate model, the frequency of firing rate oscillations in this model did not depend on the input drive but did depend on the membrane time constant (*Figure 8—figure supplement 1C*). We found that coherent oscillatory behavior survived at $n = 80$ even when the connectivity fraction $p$ was reduced down to 0.05 (i.e. only 5% of connections are made), but that oscillations broke down for smaller values of $p$ (*Figure 8—figure supplement 1A*). At $n = 10$, we found that the cutoff for stable oscillations was $p = 0.6$ (*Figure 8—figure supplement 1B*).

We next simulated the spiking model with fast and slow populations, the counterpart to our two-population rate model (*Figure 8B, C*). Choosing the membrane time constants to be the same as the rate model (i.e. 1 ms for the fast population and 10 ms for the slow population) led to issues with controllability. In particular, it led to a breakdown of coherent oscillations at intermediate driving. Referring to our results with the single-population model, we hypothesized that this was caused by the large discrepancy between the natural frequencies of the fast and slow populations. Choosing the membrane time constants to be 2 ms for the fast population and 10 ms for the slow population, we recovered frequency control. While the connectivity fraction $p$ did not affect controllability in this model, it did change the maximum frequency of the network (*Figure 8—figure supplement 1D*).

Extending our spiking model to the full eight populations of our final rate model (i.e. dividing the inhibitory population into 3 separate populations and including an excitatory population), we again observed stable oscillations with left–right alternation, head-to-tail propagation, and controllable frequency with constant phase lag. As with the rate model, we varied the strength and modularity of the excitatory connections (*Figure 8D, E*). As in the rate model, increasing excitatory connection strength increased the maximum frequency (*Figure 8D*). Rather than a loss of controllability at intermediate frequencies as in the rate model, however, the range of possible frequencies was instead reduced via an increase in the lowest frequency achieved by the network, suggesting that the fast population dominates the dynamics in the spiking network. Moreover, in contrast to the rate model, there was no change in maximum frequency upon increasing the modularity among excitatory units (*Figure 8E*), suggesting a possible saturation effect whereby the fast population already dominates the large-frequency dynamics of the network even at low degrees of modularity. Although this spiking network model responded somewhat differently than the rate models to certain parameter changes, the fact that rhythmogenesis and frequency control emerge from this model without fine-tuning supports the conclusion that network-level interactions driven by cell-type-specific connectivity patterns are sufficient for producing locomotor behavior in a more biophysically detailed model.

## Discussion

In this study, we began by postulating that cell-type-specific connectivity alone could be sufficient for producing the main phenomenological features of the spinal locomotor circuit, without requiring dynamical single-cell properties. We found that coordinated locomotion could be achieved in an inhibition-dominated network in which connectivity is determined by desired phase relationships and variable-speed control is implemented by recruitment of frequency-selective populations. Further, while structured excitatory connections were not necessary for producing coordinated locomotion or frequency control, they were useful for increasing peak locomotor frequency, albeit at the cost of losing some control at intermediate frequencies. Together, this family of models shows that network-level interactions are sufficient to generate coordinated, variable-speed locomotion. It further provides new interpretations of intersegmental excitatory and inhibitory connectivity, as well as the basic, recruitment-based mechanism of speed control.

A main conclusion of our models is the importance of intersegmental, inhibition-dominated connections for achieving coordinated locomotion, where patterns of ipsilateral and contralateral inhibition are established by desired phase relationships. Similarly, a very recent network-level model based on the mouse locomotor circuit has proposed that these are key features for obtaining coordinated locomotion in that context as well (*Komi et al., 2024*). In addition, a recent study found that patterning recurrent neural network on the locomotor circuitry of *C. elegans* produced a system that required fewer parameters and less training data than general multi-layer perceptron models for controlling a swimming agent (*Bhattasali et al., 2022*). The congruence of these results—all of which rely on emergent dynamics driven by cell-type-specific connectivity patterns to generate rhythm, without requiring complex intracellular dynamics—suggests that network-level interactions are likely key

drivers of locomotor dynamics across species and fit cross-species observations of spinal interneuron diversity. However, while these results suggest that intrinsic cellular mechanisms are not necessary to generate coordinated locomotor rhythms, they are certainly sufficient in certain circumstances, as demonstrated by previous lesion and pharmacological studies (*Grillner and Kozlov, 2021*).

Another conclusion of our models is that both projection strength and modularity among fast and slow excitatory units increase the maximum possible frequency, but at the cost of losing some control at intermediate frequencies. Recent studies in zebrafish suggest the possibility that the spinal circuit may overcome this limitation by having multiple subtypes of excitatory interneurons with different degrees of modularity. Specifically, V2a neurons with descending-only axons exhibit a greater degree of speed-dependent recruitment (suggesting a higher level of modularity), while those with bifurcating axons exhibit less (*Agha et al., 2024*). Moreover, V2a neurons with bifurcating axons fire more reliably, compared to more sparsely firing descending-only V2a neurons, and they form stronger connections to motor neurons (*Menelaou and McLean, 2019*; *Agha et al., 2024*), consistent with a hierarchical organization distinguishing interneuron rhythmogenesis from motor neuron recruitment. Including motor neurons and these distinct subtypes into a model and testing their effects will be an interesting direction for future work.

Given that the models that we have presented favor simplicity over realism, attempting to capture as much phenomenology as possible with a minimal number of tunable parameters, they are undoubtedly missing features that may be important for describing more-nuanced functional aspects of locomotion in aquatic vertebrates. For instance, we did not include in our model variations in intersegmental projection distances for dI6 and V0d neurons related to speed (*Satou et al., 2020*), nor did we include excitatory commissural interneuron classes, including V0v (*Kawano et al., 2022*) and V3 interneurons (*Böhm et al., 2022*). Moreover, other recent models of the locomotor circuit in zebrafish have highlighted the importance of electrical synapses for rapidly initiating swim bouts and characterizing early stages of development (*Roussel et al., 2020*; *Roussel et al., 2021*; *Kim and Riecke, 2023*). Including these components would be a worthwhile extension of the models presented here. Finally, investigating the ways in which the circuit-level rhythm genesis of our model might work together with and complement mechanisms based on intracellular dynamics, as assumed in classical models, will be an important direction for future work.

A somewhat unique aspect of our approach has been to develop a *hierarchy of models* to describe the same neural circuit at varying levels of detail. (See also the related approach in *Roussel et al., 2021*, which developed a series of models describing the same circuit at different stages in development.) Having such a family of models that fit within the same modeling framework enabled us to (1) use the higher-level models to better motivate the choices made in our lower-level models and (2) draw connections between models at different levels to gain additional insight into the functional roles of particular populations. Following this approach, we thus obtained greater insight into the neural mechanisms underlying behavior than would be possible from any one model individually. We expect that this general approach could be useful more broadly for characterizing neural circuits and their relation to behavior.

## Methods
### Simulation details

The multi-segment model has 30 segments and 2 sides (left and right), for a total of 60 hemi-segments. Each hemi-segment contains one unit corresponding to each neuron type. In the one-population model, there is only one inhibitory neuron type; in the two-population model, there are both fast and slow types of inhibitory neurons; in the eight-population model, there is a fast and slow type of each of excitatory, ascending ipsilateral inhibitory, descending ipsilateral inhibitory, and contralateral inhibitory neuron type.

The state of each unit is described by its firing rate as a function of time. The time course is calculated by *Equation 1*, where $h_i$ is the activity of unit $i$, and $[\cdot]_+ = \mathrm{ReLU}()$ is the rectified-linear function. The membrane time constants are set to 1 ms (10 ms) for the fast (slow) module, or 1 ms in models with no speed modules.

The numerical simulation was performed using Euler's method with timestep equal to 0.1 ms. For all simulations considered in this work, the simulation was run for 600 ms (or 6000 timesteps). The

firing rate of each unit is set to a small random rate sampled uniformly from [0, 0.01] at $t = 0$ in order to break symmetry.

For the one-population model, there were no synaptic time delays included. In the two- and eight-population models, synaptic time delays are equal to:

$$\tau_{delay} * (1 + \Delta_S), \tag{4}$$

where $\tau_{delay}$ is the base delay amount, and $\Delta_S$ is the distance (measured in number of segments) between the pair of units which the connection is between ($\Delta_S = 0$ when the units are in the same segment). For the two- and eight-population models, $\tau_{delay}$ is set to 0.2 ms for the fast population and 0.5 ms for the slow population, matching roughly the experimentally measured axonal conduction velocities detailed in *Menelaou and McLean, 2019*. Note that the synaptic time delays are determined by the identity of the source unit, not the target unit.

Units in the model are driven by a constant tonic input. This drive is varied separately for the fast and slow populations. In addition, each unit receives recurrent input from the other units according to the interneuron connectivity $W_{ij}$, which represents the strength of the connection from unit $j$ to $i$. The connectivity matrices, $W_{ij}$, are shown in *Figures 1B, 2B and 4B*.

The base value of inhibitory projections is set to –0.5. The base strength for the excitatory connections is set to $0.5 f_E$, where the multiplicative factor $f_E$ takes a value between 0 and 1 and determines the strength of excitation relative to inhibition. Except when this parameter is explicitly varied (see *Figure 6*), we set $f_E = 0.4$.

In the two- and eight-population models, we introduce the speed mixing factor, $f_{sm}$. The connection strengths for all fast-to-fast and slow-to-slow connections were multiplied by $(1 - f_{sm})$, whereas all fast-to-slow and slow-to-fast connection strengths were multiplied by $f_{sm}$. This generates a modularity of $m = 1 - 2 f_{sm}$. In the case of the eight-population model, we can apply a global speed mixing factor $f_{sm}$ or a separate speed mixing factor for the excitatory population $f_{sm,E}$ and for the inhibitory population $f_{sm,I}$. The results of varying these parameters are shown in *Figure 6*.

Ablation was introduced as an overall factor, $f_A$, that multiplied the connection strengths of all connections sourced from the ablated population. This then implies that

$$\% \text{ ablation} = (1 - f_A) \times 100\% . \tag{5}$$

## Analysis methods

The output of the simulation is a collection of time series giving the firing rate of each unit as a function of time. By inspection, it was clear that the time series settled into a sensible oscillation after an initialization period. To ensure the observations corresponded to the stable oscillatory mode, all analysis was performed on the time series from $t = 100$ ms to $t = 600$ ms (i.e. we cut out the first 100 ms).

For each unit's firing rate time series, $r(t)$, the amplitude is defined as the difference between the maximum and minimum values of the time series. That is

$$A = \max_t r(t) - \min_t r(t) . \tag{6}$$

The amplitude of the simulation is then defined as the mean amplitude across all units (or, if amplitude is reported for a particular population, across all units within that population). Errors are given by the standard deviation in amplitude across all units.

To extract the frequency of the time series $r(t)$, we calculated the period of each time series. To find the period, we considered the autocorrelation spectrum defined by:

$$C(k) = \sum_t r(t) r(t - k) \tag{7}$$

Here, $r(t) = 0$ whenever $t$ is outside the domain $t \in [100\text{ms}, 600\text{ms}]$. In all cases, the maximum autocorrelation occurs at $k = 0$. In cases with a single dominant frequency as $k$ increases away from zero, the autocorrelation drops to global minimum (which we define as $k = k_{\min}$) then rises to a local maximum at a value of $k$ equal to the period (see *Figure 3—figure supplement 2*). This is then followed by a series of local minima and maxima of lesser size. Crucially, this local maximum can be isolated as the global maximum if we only consider $k > k_{\min}$. Hence, we define the period of $r(t)$ by

$$T = \text{argmax}_{k > k_{\min}} C(k),\qquad(8)$$

and we define the frequency of $r(t)$ by $f = 1/T$. From there, frequencies of the different unit types were averaged within each hemi-segment, weighted by the amplitude. This amplitude-averaged frequency was then used to compute an unweighted mean frequency and standard deviation across all hemi-segments.

We calculate the phase $\phi$ of $r(t)$ as

$$\phi = \frac{1}{2\pi} \arg\left(\tilde{r}(\nu)\right),\qquad(9)$$

where $\arg(z)$ gives the argument (or phase) of the complex number $z$ and $\tilde{r}(\nu)$ is the Fourier coefficient of $r$ at the measured frequency, $\nu$. Notice that we have normalized our phases to be in the range $(-1/2, 1/2]$ instead of $(-\pi, \pi]$. The phase difference $\Delta\phi_{ij}$ between phases $\phi_i$ and $\phi_j$ is calculated with

$$\Delta\phi_{ij} = \frac{1}{2\pi} \arg\left(e^{2\pi i(\phi_i - \phi_j)}\right),\qquad(10)$$

to avoid erroneously large phase differences for phases near $-\frac{1}{2}$ and $\frac{1}{2}$. Each phase difference at each location is averaged via a standard amplitude-weighted mean. A circular mean and circular standard deviation are used to then average the amplitude-averaged phase differences across all hemi-segments.

## Failures of coherent oscillation

Two issues arise that can prevent the extraction of a single well-defined frequency from a completed simulation: (1) There are multiple dominant frequencies in the time series for some units, or (2) the frequencies do not agree between the fast and slow populations.

In the case of having multiple dominant frequencies, the autocorrelation spectrum no longer follows the easily interpretable shape described above. Rather, the time series and autocorrelation appear like the example in *Figure 3—figure supplement 2E, F*. A key observation is that the global minimum coincides with the first local minimum only in cases with a single dominant frequency. Therefore, to systematically find those time series with multiple dominant frequencies, we compare the frequency found using the *global* minimum as $k_{\min}$ with the frequency found using the first *local* minimum as $k_{\min}$.

In some cases (especially at high modularity), a well-defined frequency occurs within the fast population that differs significantly from a well-defined frequency within the slow population. This also demonstrates a failure for the system to oscillate coherently.

To systematically determine whether a simulation resulted in a well-defined frequency, we compare four frequency values:

1. The mean over the fast population of frequencies found using the global minimum.
2. The mean over the slow population of frequencies found using the global minimum.
3. The mean over the fast population of frequencies found using the first local minimum.
4. The mean over the slow population of frequencies found using the first local minimum.

Only if all values agree (to within a small tolerance) do we consider the frequency well-defined for the whole population.

## Spiking model implementation

The spiking models were implemented using the Brian 2 software package (*Stimberg et al., 2019*).

Each spiking model is derived from the corresponding rate model by replacing each unit of the rate model with $n$ current-based, LIF neurons. The spiking threshold was set to $V_t = 50\,\text{mV}$ and the reset potential was set to $V_r = 0\,\text{mV}$. No refractory period was set for the LIF neurons.

For a connection in the rate model from unit $i$ to unit $j$ with non-vanishing weight $W_{ij}$, a fraction $p$ of all possible connections from neurons in unit $i$ to neurons in unit $j$ are added to the spiking model. The strengths of these connections are given by

$$W_{\text{spiking}} = W_{ij}(V_t - V_r)\frac{10\,\text{ms}}{\tau_j}\frac{1}{n \cdot p}.$$

Our models employed delta function synapses, so that the currents due to recurrent connections were sums of weighted delta functions.

In place of a tonic external drive, each neuron in the spiking model receives independent EI-balanced Poisson noise. These external spikes are given a weight of $dv = \pm \frac{V_t - V_r}{10}$. The rate of the Poisson noise was chosen to promote a target firing rate in an isolated LIF neuron. Given a target rate $\nu_t$, we find the corresponding Poisson rate $\nu_p$ by numerically solving the following equation (adapted from Equation 8.54 in *Gerstner et al., 2014*):

$$\frac{1}{\nu_t} = \tau \sqrt{\pi} \int_0^x \exp\left(u^2\right) \left(1 + \text{erf}(u)\right) du \tag{11}$$

with

$$x = \frac{V_t}{dv \sqrt{2\tau \nu_p}} . \tag{12}$$

Here, erf is the error function. Target rates were chosen to be 100 Hz times the corresponding dimensionless tonic drive value.

All simulations were run with a 1-ns timestep for 600 ms total, and the spikes of every neuron were recorded.

### Detecting frequency in spiking network simulations

To extract the frequency of firing rate oscillations in the spiking models, we calculated the firing rate of each unit in 5-ms time bins by averaging the spike counts of all neurons in the unit. From these rates, an autocorrelation spectrum for each unit was calculated. When calculating the autocorrelation spectra, we discarded the initial 150 ms of the recording to avoid activity before the network settles into its stable oscillatory mode.

The rates and autocorrelation spectra for each unit are noisier than those measured in the rate model. This poses a possible challenge to our automatic frequency extraction pipeline by introducing or shifting the local minima and maxima of the autocorrelation spectra. To deal with this noise, we took an amplitude-weighted average of the autocorrelation spectrum to generate a global autocorrelation spectrum. To ensure that the shape of this spectrum is reasonable, frequencies are calculated from every peak and compared. The frequencies we report use the average of these frequency measurements. In addition, these frequencies were compared against the frequencies calculated from the fast and slow populations and using the local minimum method described above.

## Acknowledgements

We are grateful for discussions with Martha Bagnall. This work was supported by the National Institutes of Health BRAIN Initiative (U01-NS136458).

## Additional information

### Funding

| Funder | Grant reference number | Author |
| --- | --- | --- |
| National Institutes of Health | U01-NS136458 | F David Wandler<br>Benjamin K Lemberger<br>David L McLean<br>James M Murray |

The funders had no role in study design, data collection, and interpretation, or the decision to submit the work for publication.

## Author contributions
F David Wandler, James M Murray, Conceptualization, Formal analysis, Writing – original draft; Benjamin K Lemberger, Conceptualization, Formal analysis; David L McLean, Conceptualization, Writing – original draft

## Author ORCIDs
David L McLean ⬤ https://orcid.org/0000-0001-6337-2301
James M Murray ⬤ https://orcid.org/0000-0003-3706-4895

Reviewer #1 (Public review): https://doi.org/10.7554/eLife.106658.3.sa1
Reviewer #2 (Public review): https://doi.org/10.7554/eLife.106658.3.sa2
Reviewer #3 (Public review): https://doi.org/10.7554/eLife.106658.3.sa3
Author response https://doi.org/10.7554/eLife.106658.3.sa4

## Additional files

### Supplementary files
MDAR checklist

### Data availability
This is a computational study, so no data have been generated for this manuscript. Modeling code to accompany this work is available at https://github.com/fwandler/ZFswim (copy archived at *Wandler, 2025*).

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
