## [Editor Report · eLife Assessment]

In this **valuable** study, Wandler et al. provide **convincing** theoretical evidence for alternate mechanisms of rhythm generation by CPGs. Their model shows that cell-type-specific connectivity and an inhibitory drive could underlie rhythm generation. Excitatory input could act to enhance the frequency range of these rhythms. This modeling study could motivate further experimental investigation of these mechanisms to understand CPG rhythmogenesis.

---

## [Referee Report · Reviewer #1 (Public review)]

This study explores the connectivity patterns that could lead to fast and slow undulating swim patterns in larval zebrafish using a simplified theoretical framework. The authors show that a pattern of connectivity based only on inhibition is sufficient to produce realistic patterns with a single frequency. Two such networks couple with inhibition but with distinct time constants can produce a range of frequencies. Adding excitatory connections further increases the range of obtainable frequencies, albeit at the expense of sudden transitions in mid-frequency range.

Strengths:

(1) This is an eloquent approach to answering the question of how spinal locomotor circuits generate coordinated activity using a theoretical approach based on moving bump models of brain activity.

(2) The models make specific predictions on patterns of connectivity while discounting the role of connectivity strength or neuronal intrinsic properties in shaping the pattern.

(3) The models also propose that there is an important association between cell-type-specific intersegmental patterns and the recruitment of speed-selective subpopulations of interneurons.

(4) Having a hierarchy of models creates a compelling argument for explaining rhythmicity at the network level. Each model builds on the last and reveals a new perspective on how network dynamics can control rhythmicity. I liked that each model can be used to probe questions in the next/previous model.

Comments on revisions:

I am very happy to see the simplified biophysical model supporting the original findings. The authors have done an excellent job addressing my comments.

Just a small note, please change *C. elegans* to *C. elegans*.

---

## [Referee Report · Reviewer #2 (Public review)]

Summary:

The authors aimed to show that connectivity patterns within spinal circuits composed of specific excitatory and inhibitory connectivity and with varying degrees of modularity could achieve tail beats at various frequencies as well as proper left-right coordination and rostrocaudal propagation speeds.

Strengths:

The model is simple and the connectivity patterns explored are well supported by the literature

The conclusions are intuitive and support many experimental studies on zebrafish spinal circuits for swimming. The simulations provide strong support for the sufficiency of connectivity patterns to produce and control many hallmark features of swimming in zebrafish

Weaknesses:

The authors have addressed my previous concerns well. I have no further concerns.

---

## [Referee Report · Reviewer #3 (Public review)]

Summary:

Central pattern generator (CPG) circuits underly rhythmic motor behaviors. Till date, it is thought that these CPG networks are rather local and multiple CPG circuits are serially connected to allow locomotion across the entire body. Distributed CPG networks that incorporate long-range connections have not been proposed although such connectivity has been experimentally shown for several different spinal populations. In this manuscript, the authors use this existing literature on long-range spinal interneuron connectivity to build a new computational model that reproduces basic features of locomotion like left-right alternation, rostrocaudal propagation and independent control of frequency and amplitude. Interestingly, the authors show that a model solely based on inhibitory neurons can recapitulate these basic locomotor features. Excitatory sources were then added that increased the dynamic range of frequencies generated. Finally, the authors were also able to reproduce experimentally observed consequences of cell-type-specific ablations showing that local and long range, cell-type-specific connectivity could be sufficient for generating locomotion.

Strengths:

This work is novel, providing an interesting alternative of distributed CPGs to the local networks traditionally predicted. It shows cell type-specific network connectivity is as important if not more than intrinsic cell properties for rhythmogenesis and that inhibition plays a crucial role in shaping locomotor features. Given the importance of local CPGs in understanding motor control, this alternative concept will be of broad interest to the larger motor control field including invertebrate and vertebrate species.

Weaknesses:

The main weaknesses were addressed in the revision.

---

## [Author Response]

The following is the authors’ response to the original reviews.

**Reviewer #1 (Public review):**
(1)How is this simplified model representative of what is observed biologically? A bump model does not naturally produce oscillations. How would the dynamics of a rhythm generator interact with this simplistic model?

Bump models naturally produce sequential activity, and can be engineered to repeat this sequential activity periodically (Zhang, 1996; Samsonovich and McNaughton, 1997; Murray and Escola, 2017). This is the basis for the oscillatory behavior in the model presented here. As we describe in our paper, such a model is consistent with numerous neurobiological observations about cell-type-specific connectivity patterns. The reviewer is, however, correct to point out that our model does not incorporate other key neurobiological features--in particular, intracellular dynamical properties--that have been shown to play important roles in rhythm generation. Our aim in this work is to establish a circuit-level mechanism for rhythm generation, complementary to classical models that rely on intracellular dynamics for rhythm generation. Whether and how these mechanisms work together is something that we plan to explore in future work, and we have added a sentence to the Discussion to this effect.

(2) Would this theoretical construct survive being expressed in a biophysical model? It seems that it should, but even a simple biological model with the basic patterns of connectivity shown here would greatly increase confidence in the biological plausibility of the theory.

We thank the reviewer for pointing out this way to strengthen our paper. We implemented the connectivity developed in the rate models in a spiking neuron model which used EI-balanced Poisson noise as input drive. We found that we could reproduce all the main results of our analysis. In particular, with a realistic number of neurons, we observed swimming activity characterized by (i) left-right alternation, (ii) rostal-caudal propagation, and (iii) variable speed control with constant phase lag. The spiking model demonstrates that the connectivity-motif based mechanisms for rhythmogenesis that we propose are robust in a biophysical setting.

We included these results in the updated manuscript in a new Results subsection titled “Robustness in a biophysical model.”

(3) How stable is this model in its output patterns? Is it robust to noise? Does noise, in fact, smooth out the abrupt transitions in frequency in the middle range?

The newly added spiking model implementation of the network demonstrates that the core mechanisms of our models are robust to noise, since the connectivity is randomly chosen and the input drive is Poisson noise.

To test the effect of noise as it is parametrically varied, we also added noise directly to the rate models in the form of white noise input to each unit. Namely, the rate model was adapted to obey the stochastic differential equation\begin{document}$$\displaystyle  \tau_i \frac{dr_i(t)}{dt} = -r_i(t) + \left[ \sum_j W_{ij} r_j(t - \Delta_{ij}) + D_i + \sigma\xi_t \right]_+$$\end{document}

Here \begin{document}$xi_t$\end{document} is a standard Gaussian white noise and \begin{document}$\sigma$\end{document} sets the strength of the noise. We found that the swimming patterns were robust at all frequencies up to \begin{document}$\sigma = 0.05$\end{document}. Above this level, coherent oscillations started to break down for some swim frequencies. To investigate whether the noise smoothed out abrupt transitions, we swept through different values of noise and modularity of excitatory connections. The results showed very minor improvement in controllability (see figure below), but this was not significant enough to include in the manuscript.

**Author response image 1. sa4fig1:** 

(4) All figure captions are inadequate. They should have enough information for the reader to understand the figure and the point that was meant to be conveyed. For example, Figure 1 does not explain what the red dot is, what is black, what is white, or what the gradations of gray are. Or even if this is a representative connectivity of one node, or if this shows all the connections? The authors should not leave the reader guessing.

All figure captions have been updated to enhance clarity and address these concerns.

**Reviewer #2 (Public review):**
(1) Figure 1A, if I interpret Figure 1B correctly, should there not be long descending projections as well that don't seem to be illustrated?

Thank you for highlighting this potential point of confusion. The diagram in question was only intended to be a rough schematic of the types of connections present in the model. We have added additional descending connections as requested

(2)Page 5, It would be good to define what is meant by slow and fast here, as this definition changes with age in zebrafish (what developmental age)?

We have updated the manuscript to include the sentence: “These values were chosen to coincide with observed ranges from larval zebrafish.” with appropriate citation.

**Reviewer #3 (Public review):**
(1) The authors describe a single unit as a neuron, be it excitatory or inhibitory, and the output of the simulation is the firing rate of these neurons. Experimentally and in other modeling studies, motor neurons are incorporated in the model, and the output of the network is based on motor neuron firing rate, not the interneurons themselves. Why did the authors choose to build the model this way?

We chose to leave out the motor neurons from our models for a few reasons. While motor neurons read out the rhythmic activity generated by the interneurons and may provide some feedback, they are not required for rhythmogenesis. In fact, interneuron activity (especially in the excitatory V2a neurons (Agha et al., 2024)) is highly correlated with the ventral root bursts within the same segment. This suggests that motor neurons are primarily a local readout of the rhythmic activity of interneurons; therefore, the rhythmic swimming activity can be deduced directly from the interneurons themselves.

Moreover, there is a lack of experimental observation of the connectivity between all the cell types considered in our model and motor neurons. Hence, it was unclear how we should include them in the model. To address this, we are currently developing a data-driven approach that will determine the proper connectivity between the motor neurons and the interneurons, including intrasegmental connections.

(2) In the single population model (Figure 1), the authors use ipsilateral inhibitory connections that are long-range in an ascending direction. Experimentally, these connections have been shown to be local, while long-range ipsilateral connections have been shown to be descending. What were the reasons the authors chose this connectivity? Do the authors think local ascending inhibitions contribute to rostrocaudal propagation, and how?

The long-range ascending ipsilateral inhibitory connections arises from a limitation of our modeling framework. The V1 neurons that provide these connections have been shown experimentally to fire later than other neurons (especially descending V2a neurons) within the same hemisegment (Jay et al., J Neurosci, 2023); however, our model can only produce synchronized local activity. Hence, we replace local phase offsets with spatial offsets to produce correctly structured recurrent phasic inputs. We are currently investigating a data-driven method for determining intrasegmental connectivity which should be able to produce the local phase offset and address this concern; however, this is beyond the scope of the current paper.

(3) In the two-population model, the authors show independent control of frequency and rhythm, as has been reported experimentally. However, in these previous experimental studies, frequency and amplitude are regulated by different neurons, suggesting different networks dedicated to frequency and amplitude control. However, in the current model, the same population with the same connections can contribute to frequency or amplitude depending on relative tonic drive. Can the authors please address these differences either by changes in the model or by adding to the Discussion?

Our prior experimental results that suggested a separation of frequency and amplitude control circuits focus on motor neuron recruitment, instead of interneuron activity (Jay et al., J Neurosci 2023; Menelaou and McLean, Nat Commun 2019). To avoid potential confusion about amplitudes of interneurons vs. of motor neurons, we have removed the results from Figure 3 about control of amplitude in the 2-population model, instead focusing this figure on the control of frequency via speed-module recruitment. For the same reason, we have removed the panel showing the effects of targeted ablations on interneuron amplitudes in Figure 7. We have kept the result about amplitude control in our Supplemental Figure S2 for the 8-population model, but we try to make it clear in the text that any relationship between interneuron amplitude and motor neuron amplitude would depend on how motor neurons are modeled, which we do not pursue in this work.

(4) It would be helpful to add a paragraph in the Discussion on how these results could be applicable to other model systems beyond zebrafish. Cell intrinsic rhythmogenesis is a popular concept in the field, and these results show an interesting and novel alternative. It would help to know if there is any experimental evidence suggesting such network-based propagation in other systems, invertebrates, or vertebrates.

We have expanded a paragraph in the Discussion to address these questions. In particular, we highlight how a recent study of mouse locomotor circuits produced a model with similar key features (Komi et al., 2024). These authors made direct use of experimentally determined connectivity structure and cell-type distributions, which informed a model that produced purely network-based rhythmogenesis. We also point out that inhibition-dominated connectivity has been used for understanding oscillatory behavior in neural circuits outside the context of motor control (Zhang, 1996; Samsonovich and McNaughton, 1997; Murray and Escola, 2017). Finally, we address a study that used the cell-type specific connectivity within the *C. elegans* locomotor circuit as the architecture for an artificial motor control system and found that the resulting system could more efficiently learn motor control tasks than general machine learning architectures (Bhattasali et al. 2022). Like our model, the Komi et al. and Bhattasali et al. models generate rhythm via structured connectivity motifs rather than via intracellular dynamical properties, suggesting that these may be a key mechanism underlying locomotion across species.

**Reviewer #1 (Recommendations for the authors):**
(1) Express this modeling construct in a simple biophysical model.

See the new Results subsection titled “Robustness in a biophysical model.”

(2) Please cite the classic models of Kopell, Ermentrout, Williams, Sigvardt etc., especially where you say "classic models".

We have added relevant citations including the mentioned authors.

(3) "Rhythmogenesis remain incompletely understood" changed to "Rhythmogenesis remains incompletely understood".

We chose not to make this change since the ‘remain’ refers to the plural ‘core mechanisms’ not the singular ‘rhythmogenesis’.

**Reviewer #3 (Recommendations for the authors):**
(1) The figures are well made; however, it would help to add more details to the figure legends. For example, what neuron's firing rate is shown in Figure 1C? What is the red dot in 1B? Figures 3E,F,G: what is being plotted? Mean and SD? Blue dot in Figure 5C?

All figure captions have been updated to enhance clarity and address these concerns.

(2) A, B text missing in Figure 7.

We have revised this figure and its caption; please see our response to Comment 3 above.

(3) It would be nice to see the tonic drive pattern that is fed to the model for each case, along with the different firing rates in the figures. It would help understand how the tonic drive is changed to rhythmic activity.

The tonic drive in the rate models is implemented as a constant excitatory input that is uniform across all units within the same speed-population. There is no patterning in time or location to this drive.

References

(1) Moneeza A Agha, Sandeep Kishore, and David L McLean. Cell-type-specific origins of locomotor rhythmicity at different speeds in larval zebrafish. eLife, July 2024

(2) Nikhil Bhattasali, Anthony M Zador, and Tatiana Engel. Neural circuit architectural priors for embodied control. In S. Koyejo, S. Mohamed, A. Agarwal, D. Belgrave, K. Cho, and A. Oh, editors, Advances in Neural Information Processing Systems, volume 35, pages 12744–12759. Curran Associates, Inc, 2022.

(3) Salif Komi, August Winther, Grace A. Houser, Roar Jakob Sørensen, Silas Dalum Larsen, Madelaine C. Adamssom Bonfils, Guanghui Li, and Rune W. Berg. Spatial and network principles behind neural generation of locomotion. bioRxiv, 2024

(4) James M Murray and G Sean Escola. Learning multiple variable-speed sequences in striatum via cortical tutoring. eLife, 6:e26084, May 2017.

(5) Alexei Samsonovich and Bruce L McNaughton. Path integration and cognitive mapping in a continuous attractor neural network model. Journal of Neuroscience, 17(15):5900–5920, 1997.

(6) K Zhang. Representation of spatial orientation by the intrinsic dynamics of the head-direction cell ensemble: a theory. Journal of Neuroscience, 16(6):2112–2126, 1996.